# Progressive Unspecified Motor Speech Disorder: A Longitudinal Single Case Study of an Older Subject

**DOI:** 10.3390/geriatrics7030052

**Published:** 2022-04-24

**Authors:** Benedetta Basagni, Sonia Martelli, Livia Ruffini, Anna Mazzucchi, Francesca Cecchi

**Affiliations:** 1IRCCS Fondazione Don Carlo Gnocchi-ONLUS, Via di Scandicci, 269, 50143 Florence, Italy; annamazzucchi@gmail.com (A.M.); fcecchi@dongnocchi.it (F.C.); 2IDIPSI, Istituto di Psicoterapia Sistemica Integrata, Strada Vallazza, 6, 43100 Parma, Italy; sonitza@hotmail.com; 3Nuclear Medicine Division, Azienda Ospedaliero-Universitaria di Parma, 43126 Parma, Italy; liviaruffini@gmail.com

**Keywords:** progressive speech disorder, neuroradiological assessment, PET, cognitive decline

## Abstract

Introduction: In a few cases, neurodegenerative diseases debut with a speech disorder whose differential diagnosis can be difficult. Case Report: We describe the case of a right-handed woman with a progressive speech impairment, which debuted when she was 80 years old. We report the results of neurological, neuropsychological, and imaging assessments with positron emission tomography (PET) over a period of nine years. Metabolic PET with 18F-FDG was performed at the age of 81 and repeated two years later due to the worsening of symptoms; amyloid PET with 18F-flutemetamol was performed at the age of 86. All PET results were quantitatively analyzed. A speech impairment remained the isolated neurological symptom for a long time, together with a mood disorder. Early FDG-PET showed hypometabolism in the left superior and inferior frontal areas, in the left superior temporal area, and in the right superior frontal area. Two years later, the hypometabolic area was more extensive. Amyloid PET was qualitatively and quantitatively normal. Nine years after the first symptoms, the speech production progressively worsened until complete anarthria, in association with writing impairment onset and signs of behavioral impairments. No signs of motor involvement were found. Conclusions: A progressive articulatory disorder without an evolution of motor disorders may be a distinct neurological degenerative entity, mainly affecting speech production for very a long time and with a specific early metabolic pattern in brain FDG-PET in the language production area. Monitoring patients with FDG-PET could predict the disease evolution years before a clinical deterioration.

## 1. Introduction

The first symptoms of neurodegenerative diseases can affect different components, mostly concerning the motor system and cognitive functions. More infrequently, the onset of the pathology concerns verbal production, in different modes. Forms of primary progressive aphasia have been described [1]. However, in a few cases the articulatory disturbance is not accompanied by aphasic alterations.

This is the case for patients with a neurodegenerative disorder that begins focally with spastic dysarthria, which has been labeled as “progressive spastic dysarthria” [2]. The neurological evolution of these forms mostly involves the motor and premotor cortex as well as the descending corticospinal and corticobulbar pathways (as primary lateral sclerosis or anterior opercular syndrome, also known as Foix–Chavany–Marie syndrome). Nevertheless, a few other works have described patients with isolated dysarthria at the onset that evolved into fronto-temporal dementia [3,4,5] studied nine patients in which progressive dysarthria remained the sole neurological sign for two years from the onset. The symptoms of the patients evolved into corticobasal degeneration, motor neuron disease dementia, amyotrophic lateral sclerosis, and amyotrophic lateral sclerosis aphasia. Considering that, for two of these patients, progressive dysarthria remained the only neurological sign at the latest examination, the authors concluded that progressive dysarthria may also be considered to be a distinct entity.

Other forms with a selective onset of the articulatory component of language concern the diagnosis of primary progressive apraxia of speech (PP-AOS). Apraxia of speech is a disorder of speech motor planning or programming, which may be present without facial apraxia and aphasia. Hence, although articulatory apraxia is often a symptom that can occur together with progressive aphasia, it can also occur in isolation. AOS may be the initial symptom of neurodegenerative diseases in the absence of aphasia or other motor dysfunctions [6]. The verbal production is characterized by a slow rate, articulatory distortions, and distorted sound substitutions. Articulatory groping and trial and error articulatory movements are frequently evident [7].

In this paper, we describe the case of a patient who debuted with a pure speech disorder in the absence of a cognitive or motor decline for several years. Over nine years we followed the evolution of the symptoms. We report the PET imaging, performed over time to assess the glucose consumption and amyloid deposits, and discuss the results within the context of the differential diagnosis between progressive spastic dysarthria and primary progressive apraxia of speech.

## 2. Case Report

The patient was a right-handed woman with 15 years of education. She retired after a long career as an Italian–English interpreter.

In 2012, aged 80, she passed her first neurological and neuropsychological assessment. She complained about generic difficulties in speech articulation. The disturbance was then essentially subjective where the listener only partially detected anomalies. Both the neurological assessment and neuropsychological tests were negative. At that time, a magnetic resonance (MR) test was also prescribed, which showed a mild dilation of the sulci of the bilateral front parietal convexities. Small areas of altered signals at the level of the periventricular white matter, which were probably of a vascular nature, were also highlighted.

In 2013, the patient showed a mild worsening of symptoms with a limited objective articulation difficulty. Hence, the patient underwent brain PET with 18F-FDG, according to the procedure previously described by Mora et al., 2016 [8]. The brain PET/CT recording started 30 min after a tracer injection (185 MBq) on a 3D PET/CT Discovery ST (GE Healthcare, Chicago, IL, USA). The quantitative analysis of the PET images was performed using SPM5 software (Wellcome Trust Centre for Neuroimaging, London, UK, http://www.fil.ion.ucl.ac.uk/spm (accessed on 10 January 2013) implemented in MATLAB R2014a [9]. The PET dataset was spatially normalized using the SPM5 PET template and smoothed with a Gaussian filter of 8 mm at FWHM (Figure 1).

Statistical parametric mapping (SPM) was carried out to identify the areas of significant hypometabolism. The results were displayed in the Talairach Atlas [10]. The hypometabolic areas were investigated at a very high level of significance (*p* = 0.0001 and *p* = 0.05). In 2013, early FDG-PET showed hypometabolism in the left inferior frontal gyrus (BA 45), in the prefrontal cortex (BA 9 and BA 10 in the left hemisphere, BA 10 in the right one), and in the left superior temporal gyrus (Figure 2).

In 2014, the patient passed three neurological assessments, which were still negative. The dysarthric disorder was, by this time, recognizable by the listener and was labeled as “spastic”. The patient began verbal communication, but it was frequently interrupted in correspondence with the first phonemes (consonants) with the inability to restart speech production and the physical manifestation of impatience. The naming task of the Aachener Aphasia Test [11] was also administered and did not highlight any lexical access impairments. Signs of depression were detected; an antidepressant therapy was prescribed. The patient did not undergo speech therapy.

In 2015, due to the progressively slow worsening of the articulation symptoms, the patient was submitted once again for brain PET with 18F-FDG. The hypometabolism was more extensive and involved the left middle and superior temporal gyrus (BA 42, BA 21, and BA 38), the superior frontal gyrus bilaterally, the left inferior frontal gyrus (BA 9 and BA 46), and the anterior cingulate bilaterally. Moreover, hypometabolism was present bilaterally in the subcortical structures as caudate and thalamus (Figure 3).

In 2017, amyloid PET with 18F-flutemetamol was performed to exclude a β-amyloid deposition.

The PET scan was performed using a whole-body hybrid system Discovery IQ (GE Healthcare) operating in a three-dimensional detection mode. A head holder was used to restrict patient movements. The cerebral emission scan began 90 min after a slow i.v. bolus injection of 18F-flutemetamol (185 MBq/kg). The PET data were acquired over 20 min (4 frames of 5 min each). A low-dose CT scan of the head was performed before the emission scan for attenuation corrections. All PET sinograms were reconstructed with a 3D maximum likelihood ordered subset expectation maximization (3D OSEM) algorithm (filter cut off, 5.00 mm; subsets, 12; iterations, 4; Z axis filter, standard; matrix size, 256 × 256; and DFOV, 30 cm) that included corrections for scatter, random events, dead time, and attenuation (from the CT).

The PET images were visually assessed by two trained (https://www.readvizamyl.com/ (accessed on 15 November 2015)) independent readers who were blinded to each other, using a previously described technique [12]. The images were reviewed in color using a spectrum color scale (Figure 4).

A regional quantification of the 18F-flutemetamol uptake was performed using a fully automated PET-only method, as previously described by Thurfjell et al. [13]. This technique was based on the categorization of the scans using a composite standardized uptake value ratio (SUVR) threshold derived from an autopsy cohort. The SUVRs in the cerebral cortex were automatically generated and normalized to the pons using CortexID Suite software (http://www3.gehealthcare.com/en/products/categories/advanced_visualization/applications/cortexid) (accessed on 25 February 2022), which created a z-score for each of the examined cerebral areas. The z-score defined the number of SDs above the normal mean obtained from the intrinsic software database control group (>100 amyloid-negative healthy controls). A z-score > 2.0 indicated an abnormally increased regional amyloid burden, corresponding with a composite SUVR of 0.59, which provided a 99.4% concordance with the visual assessment [13]. In this study case, a normal tracer distribution at the qualitative analysis with a z-score less than 2.0 in all the examined regions was identified with amyloid-negative PET (Figure 5).

In the same year, the neurological assessment remained negative except for a speech articulation disorder as an isolated symptom. A neuropsychological assessment was also administered. WAIS-IV subtests showed results above the normative data. No signs of dementia were revealed. The speech production worsened and was described as a further articulatory slowdown.

In 2021, the neurological assessment showed a further severe worsening of the phono-articulation capacities of the subject. The speech was no more intelligible and the patient mostly expressed herself by writing.

A broad neuropsychological assessment was conducted at that point (Table 1). Due to difficulties in oral expressions, a few tests were proposed that allowed the patient to provide the answer in a written form. The results showed the presence of bucco-facial apraxia, but limb ideomotor apraxia was absent. Constructional apraxia was also present, together with a working memory and sustained attention disease. Verbal comprehension of simple orders, written naming, gestural perception, semantic retrograde memory, and logical deductive reasoning were adequate. With respect to writing, although possible and generally intelligible, a few errors in graphemic selection and sequencing started to be present.

The assessment of the functioning of the subject in the activities of daily living was conducted by interviewing her cohabiting sister. This revealed that the patient was autonomous in nutrition, hygiene, and locomotion although there was occasional urinary incontinence. There were no difficulties in recognizing people and the memory and temporal-spatial orientation appeared to be preserved. The sister, on the other hand, reported a change in behavioral aspects and character. Dysfunctional behaviors such as outbursts of anger, psycho-motor agitation, and episodes of aggression on objects and people were reported. A reduction in spontaneous activity was also present. However, no episodes of nocturnal confusion, inappetence/hyperphagia, sensory hallucinations, delusions and confabulations, or sexual disinhibition were reported. The mood was strongly depressed and not compensated by drug therapy.

A further PET scan was planned, but in the weeks following the neuropsychological assessment, the patient showed a sudden and severe worsening of the global conditions (writing ability lost, deep depression, enticement, inertia, apathy, and refusal of food) and the scan was not performed. Table 2 resumes the global timeline of the neurological, neuropsychological, and imaging assessments.

## 3. Discussion

We present the clinical history of a patient with a neurological disorder in which a phono-articulation impairment remained the sole neurological symptom over nine years. No signs of body motor involvement were found. Writing anomalies and a mild cognitive impairment appeared nine years after the symptom onset, at the age of 89. Shortly after the onset of these symptoms, the global clinical condition of the patient dramatically worsened.

Although a few sporadic cases in which speech disorders remain the isolated neurological findings for years are described in the literature, in the majority of cases the disturbance evolves into motor/premotor cortex and descending corticospinal/corticobulbar pathway impairments. This was not the case with our subject. Only nine years after the onset, the speech disorder was complete and the clinical conditions dramatically precipitated in a few weeks. At that point, the cognitive decline became evident (e.g., the writing quickly became unintelligible) and the mood disorder worsened with a refusal to collaborate and interact.

At the beginning, such a pattern could have been easily confounded with a psychogenic cause because no other neurological signs were detectable, and mood disorders often together occur alongside. Psychogenic forms of speech disorders are widely described in the literature [14,15]. When the disturbance manifested, our main diagnostic hypotheses concerned progressive dysarthria and progressive articulatory apraxia.

With respect to the neuroradiological data, the clinical picture began with hypometabolism in the left superior and inferior frontal areas, in the left superior temporal area, and in the right superior frontal area and then more extensively involved the temporal and frontal lobes as well as in the right anterior cingulate and in the caudate and thalamus bilaterally.

The literature data based on functional imaging studies suggest that proper articulation during speech output is a bilateral process involving bidirectional motor, sensory, and supramodal integrations [16].

In our patient, early metabolic imaging with FDG-PET revealed hypometabolism in the right and left premotor cortex, which is associated with motor programming and articulatory coding [16,17,18]. Hypometabolism was also detected in Broca’s complex (BA 45) [19] as well as in the language comprehension area (BA 22) in the temporal lobe [20].

A late PET assessment three years after the symptom onset revealed a greater extension of hypometabolism with the involvement of BA 6, basal ganglia, and thalamus, which are considered to be part of the language production system [21]. Moreover, hypometabolism was demonstrated at the late PET scan in BA 46, which is considered to be a major frontal lobe executive functioning area [22].

The late FDG-PET also showed the involvement of the language reception/understanding system [21]: BA 21 and BA 42 in the core Wernicke’s area are involved in word recognition and BA 38 is considered to be a fringe of the Wernicke’s area [21].

Hypometabolism was also demonstrated in the anterior cingulate; previous fMRI studies have suggested this is likely involved in the attentional control necessary for word production [18,23].

Taken together, the metabolic PET data confirm the integrated network of the cortical and subcortical areas in speech production. In particular, the association of hypometabolism in the bilateral premotor cortex and in Broca’s complex is considered to be a key node in the transformation of information across the cortical networks involved in speech production [24]. This could be a distinctive feature both of progressive dysarthria and of apraxia of speech.

Regarding the ACC involvement, it was recently reported that the anterior cingulate cortex showed the most aging-related brain metabolic dysfunctions, correlating with decreasing executive processing in otherwise healthy, cognitively intact, and amyloid-negative volunteers [25]. The authors indicated for the first time that ACC metabolism is a mediator of the relationship between age and fluency even in older cognitively normal adults free of amyloids.

In our patient, ACC hypometabolism appeared at the second FDG-PET evaluation, three years after the symptom onset; this could be referred to as “cognitive aging” [26].

Moreover, the ACC may also be involved in symptomatic depressed states, as in our patient who showed a decreased activity at FDG-PET [27].

Brain SPECT perfusions and PET metabolism appear to be discordant biomarkers in depressive disorders, as recently reported [28], with no significant relationship between brain metabolism and chronicity of illness.

A few qualitative characteristics of the disturbance observed in our patient supported the diagnostic hypothesis of PP-AOS, such as a lack of dysphonia and hypernasality. The presence of bucco-facial apraxia and the subsequent onset of aphasic symptoms (verbal comprehension and a writing deficit) also supported the hypothesis of a long phase of disorders limited to articulatory apraxia, only latterly evolving toward aphasia. Qualitatively, however, the disorder seemed to have the characteristics of spastic dysarthria.

This study had a few limitations. First of all was the lack of quantitative assessments of a speech disorder over time. The case was investigated retrospectively and not all data were found. Similarly, the study lacked a T1-MRI scan that might have been important to evaluate the degree of atrophy of the patient at different timepoints. Finally, the FDG-PET scans were quantitatively assessed with SPM version 5, which was not the most up-to-date version; however, the intention of the authors was to accurately report the patient data in a rigorous way as they were produced during years without reprocessing.

The case presented in this report confirmed the hypothesis that articulatory disease without an evolution of a motor disorder may be a distinct neurological degenerative entity that affects sole speech production for a long time.

## 4. Conclusions

The case we presented demonstrated how a speech disorder can be selective and limited for a long time, suggesting caution in cases in which patients subjectively complain of alterations in phono-articulation before concluding a psychogenic disorder. Furthermore, an early assessment of brain glucose hypometabolism with PET could be predictive of the evolution of symptoms and its execution could be useful for a differential diagnosis.

## Figures and Tables

**Figure 1 geriatrics-07-00052-f001:**
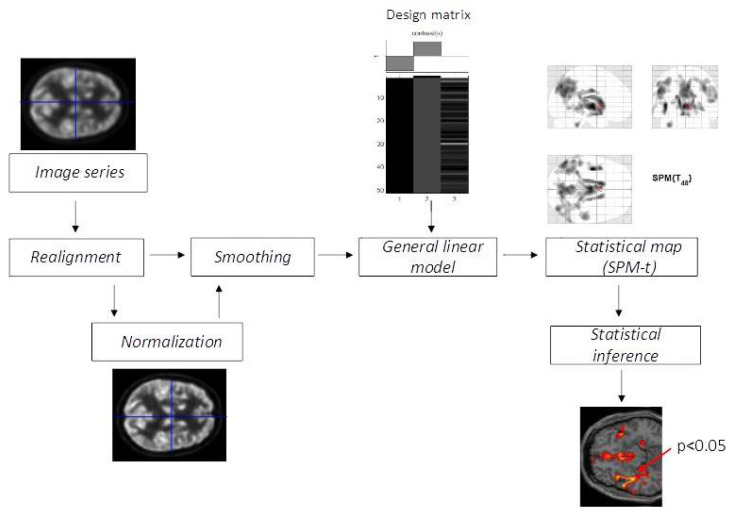
Flow chart of FDG-PET image pre-processing. The colour-coded areas (red arrow) indicate the locations where the voxel values of a patient are significantly different from the normal control group.

**Figure 2 geriatrics-07-00052-f002:**
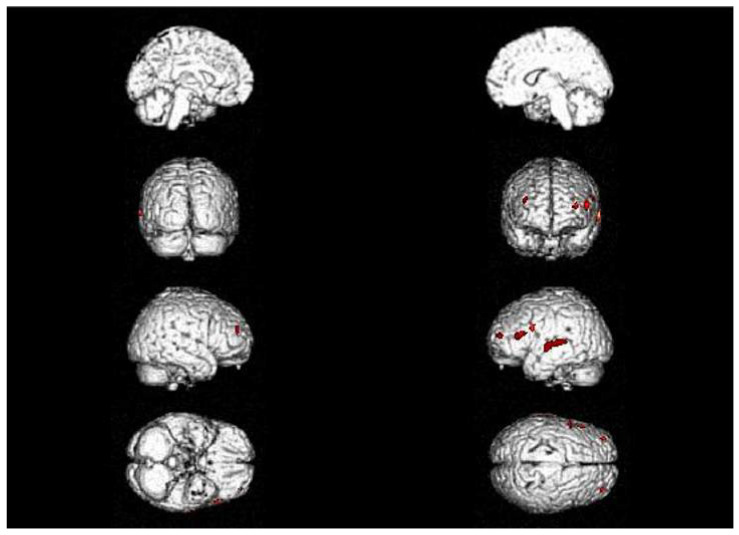
Brain PET with 18F-FDG one year (2013) after symptom onset (early FDG-PET). SPM map of significant hypometabolic areas (*p* = 0.0001) in the left inferior frontal gyrus (BA 45), in the prefrontal cortex (BA 9 and BA 10 in the left hemisphere, BA 10 in the right one), and in the left superior temporal gyrus.
The results are displayed
on the structural MR template included in SPM5.

**Figure 3 geriatrics-07-00052-f003:**
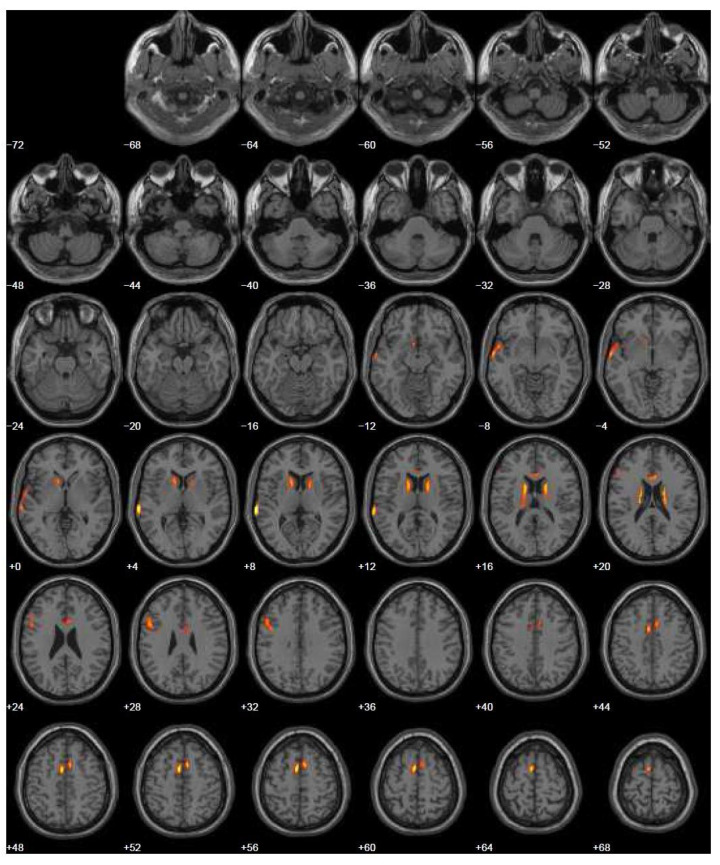
Brain PET with 18F-FDG three years after symptom onset (2015). SPM map of significant hypometabolic areas (*p* = 0.05) overlaid on the structural MR template included in SPM5. Hypometabolism was detected in left middle and superior temporal gyrus (BA 42, BA 21, and BA 38), superior frontal gyrus bilaterally, left inferior frontal gyrus (BA 9 and BA 46), anterior cingulate bilaterally, and subcortical structures bilaterally.

**Figure 4 geriatrics-07-00052-f004:**
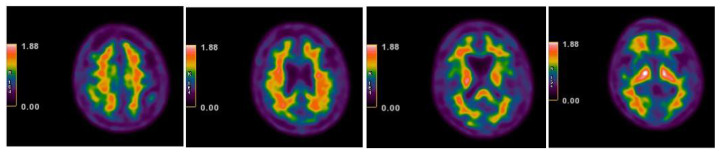
Amyloid-negative PET scan with 18F-flutemetamol five years after symptom onset: SUM PET representative axial slices on the spectrum color scale (normal tracer distribution).

**Figure 5 geriatrics-07-00052-f005:**
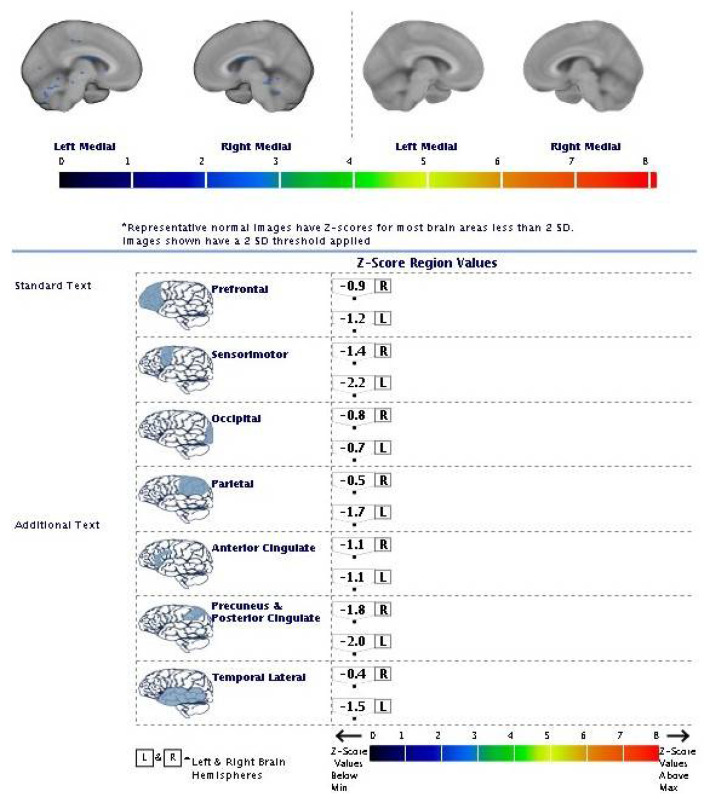
Regional quantification of the 18F-flutemetamol uptake five years after symptom onset: z-score images (cut-off value +2 SD).

**Table 1 geriatrics-07-00052-t001:** Results of neuropsychological assessment in 2021. According to the equivalent score method, raw scores were classed into five ranges corresponding with five categories (0, 1, 2, 3, and 4). We considered pathologic an equivalent score of 0, which indicated a performance corresponding with the worst 5% of the normative sample.

TEST	Raw Score	Results
Bucco-facial apraxia (Spinnler and Tognoni, 1987)	6/20	Pathological
Ideomotor apraxia (Spinnler and Tognoni, 1987)	19/20	Normal
Token test (Spinnler and Tognoni, 1987)	13/36	Pathological
CPM47 (Caltagirone et al., 1995)	23/36	Normal
ACE-R (Siciliano et al., 2016) Naming	9/10	Normal
Comprehension (written)	1/1	Normal
Three-stage order	3/3	Normal
Images	3/3	Normal
Writing	0/1	Pathological
Retrograde memory	4/4	Normal
Visuo-spatial abilities Pentagon	0/1	Pathological
Cube	1/1	Normal
Clock	1/5	Pathological
Perceptive abilities Points	1/4	Pathological
Letters	4/4	Normal

**Table 2 geriatrics-07-00052-t002:** Timeline of neurological, neuropsychological, and neuroradiological assessments.

2012	Neuropsychological assessment RMN	Negative Mild dilation of the sulci of the bilateral front parietal convexities
2013	FDG-PET	Hypometabolism in the right and left premotor cortex
2014	Three neurological assessments and lexical tests	Mild isolated motor speech disorder
2015	Neurological assessment FDG-PET	Mild isolated motor speech disorder hypometabolism; more extensive involving left temporal gyrus, frontal gyrus bilaterally, cingulate, caudate, and thalamus bilaterally
2017	Neurological assessment Neuropsychological assessment (WAIS-IV) Speech therapist assessment Amyloid PET	Mild isolated motor speech disorder No abnormal β-amyloid deposits
2021	Neurological assessment Broad neuropsychological assessment	Mild cognitive impairment Sudden worsening of the condition of the patient

## Data Availability

The data can be required from the corresponding author.

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
