# Peer review of "Progressive Unspecified Motor Speech Disorder: A Longitudinal Single Case Study of an Older Subject"

_geriatrics, 2022, doi:10.3390/geriatrics7030052_

Round 1
Reviewer 1 Report
The authors present a highly interesting case of an elderly woman who presents for several years with an isolated progressive motor speech disorder. They present relatively detailed imaging findings, general neurologic and detailed neuropsych findings (when available). Unfortunately, they provide almost no description of the actual motor speech disorder. Moreover, the details they do provide do not support the diagnosis of spastic dysarthria, but instead are more consistent with primary progressive apraxia of speech (PPAOS). I will detail below how I come to this conclusion. My only other concern is English language editing.
Line 33. In most dysarthrias it is the nervous system, not the muscles, that are impaired. I wonder if this could be reworded to "...in which muscle control for speech is damaged, paralyzed..."
Line 42. The word "nature" here doesn't seem to be the correct choice as it implies an intrinsic characteristic of the dysarthria, which should be known. Perhaps what is meant here is "etiology" or "cause"?
Line 64 & 70. You indicate that the presenting complaint for the first 2 years is articulation. This was my first hint that the problem might be PPAOS rather than spastic dysarthria. In my experience, spastic dysarthria often presents first with voice changes, although I acknowledge there are exceptions. However, isolated articulation complaints are pathognomonic for PPAOS.
Line 90. You indicate that the speech disorder is now recognizable as spastic. However, no methodology for assessing speech production is provided and no description of speech features are provided for reaching the diagnosis of spastic dysarthria. The description that IS provided (lines 91-92) are once again not typical of spastic dysarthria but very suggestive of AOS.
Line 96. Once again, no method of assessing speech and no description of speech features for the 2015 findings.
Line 117. Once again, no method of assessing speech. The only description of speech suggests slowing and articulation impairment, again consistent with AOS, but no features that would otherwise suggest spastic dysarthria, especially now that it is described as "worse." At this point, if this were spastic dysarthria we would expect there to be significant dysphonia and hypernasality, but this is not mentioned at all.
Line 123. Buccofacial apraxia commonly accompanies AOS but does not commonly co-occur with spastic dysarthria unless the spastic dysarthria also co-occurs with AOS.
Line 125. Given 13/36 on the Token Test, a statement of impaired verbal comprehension should be added. That observation, along with writing deficits, could be the first indication of aphasia, that can appear late in PPAOS (see references I have included below).
Table 2. What is "soft" isolated dysarthria? Is "mild" the better term here? Again, I would ask you to reconsider this diagnosis.
Lines 199-200. I see that this case was conducted retrospectively so I presume no speech recordings are available. It would be great for readers to listen to this patient and perhaps debate the diagnosis.
Can the authors comment on whether this patient received any speech therapy throughout the course of her progression? If so, are there any details to the nature of this therapy?
I am least qualified to critique to discussion of imaging findings, and indeed these seem to be the greatest strength of the case study, as they provide the most reliably reported data. Again, I would ask the authors to reconsider how they interpret the networks in light of how these same brain areas support motor planning and programming in addition to motor execution. This should then be tied back to the patient's impairments.
I hope the authors can provide more information to support the diagnosis of spastic dysarthria for this patient. If not, I hope they will review the PPAOS literature (only a brief sample cited below) and consider this an alternative, and I would argue better, more fitting diagnosis (as least as their patient is currently described in this paper). At a minimum, the language would need to be changed to indicate the patient had an unspecified motor speech disorder, because the current description does not support the diagnosis of spastic dysarthria.
Finally, in spite of my disagreement with the diagnosis (which may stem entirely from incomplete patient description), I congratulate the authors on an important contribution to the literature. Patients with isolated communication disorders often go ignored and I am gratified that this patient had such a thorough medical workup.
Josephs, K. A., Duffy, J. R., Strand, E. A., Machulda, M. M.,
Senjem, M. L., Master, A. V., . . . Whitwell, J. L. (2012). Characterizing
a neurodegenerative syndrome: Primary progressive
apraxia of speech. Brain, 135, 1522–1536.
Josephs, K.A., Duffy, J.R., Clark, H.M. et al. A molecular pathology, neurobiology, biochemical, genetic and neuroimaging study of progressive apraxia of speech. Nat Commun 12, 3452 (2021). https://doi.org/10.1038/s41467-021-23687-8
Utianski RL, Duffy JR, Clark HM, Strand EA, Botha H, Schwarz CG, Machulda MM, Senjem ML, Spychalla AJ, Jack CR Jr, Petersen RC, Lowe VJ, Whitwell JL, Josephs KA. Prosodic and phonetic subtypes of primary progressive apraxia of speech. Brain Lang. 2018 Sep;184:54-65. doi: 10.1016/j.bandl.2018.06.004. Epub 2018 Jul 4. Erratum in: Brain Lang. 2020 Jun;205:104792. PMID: 29980072; PMCID: PMC6171111.
Reviewer 2 Report
Basagni and colleagues discuss the disease course of a 80 years old patient presenting with progressive dysarthria without motor impairment over a course of 9 years. Hypometabolic patterns are discussed and indicate left-lateralised premotor cortex and inferior frontal cortex involvement as well as superior temporal hypometabolism. No post mortem validation of the underlying pathology is discussed. Positive here is that extensive data are available for this patient which is definitely very valuable and unique, however the presentation and interpretation of results can be improved.
Main points:
Data analyses
The hypometabolic patterns on FDG PET have been analysed by SPM5 and are discussed on two figures: why did the authors use outdated software (SPM5 vs SPM12)? Which specific statistical analysis was performed in SPM: Was this based on a voxel-wise one-sample t test?
Can the statistical threshold be clarified in more detail: (p=0.0001, p=0.05) does this refer to Poline et al., ? For voxelwise analyses, a voxel-level uncorrected p < 0.001 combined with cluster-level family-wise error (FWE)-corrected p <0.05 (Poline et al., 1997) is standard and robust.
SPM normalizes to MNI space: why did the authors choose to report in Talairach space?
The Amyloid PET is shown in figure 3: is this a SUM PET image or a SUVR image? Can the authors add a scale bar to the figure?
What was the acquisition period of the amyloid PET scan (90min-…?)
How were amyloid data processed?
Results
Figure 1 and 2: A comparison between the different timepoints would be facilitated by showing the clusters overlaid on the same template brain and present these in two panels of the same figure.
Table 1: can the column with “range” be removed and the total of each score just added after each test (e.g. ACE-R (/10))?
Results are normal or pathological based on which normative measures (independent cohort of normal, calculation of z-score, ..)? Is it possible to add a Z-score to show the exact level of abnormality instead of a label normal/pathological?
Discussion
One mentions that early hypometabolic patterns on FDG PET are seen in “the right and left premotor cortex”, however I would emphasize the left-lateralised involvement first, which seems more extensive in the early disease course, consistent with the language phenotype.
Moreover, the hypometabolic patterns on FDG PET resemble those seen in the nonfluent variant of primary progressive aphasia (nfvPPA) (premotor cortex, inferior frontal) (see Josephs and Whitwell papers on imaging biomarkers). Hence, this is not a unique distinguishing feature. Could the authors discuss this overlap and how a clinician might distinguish these two phenotypes?
Not only premotor cortex, supplementary motor area and ACC, but also basal ganglia are often affected by tauopathy in nfvPPA (E.g. see https://doi.org/10.1007/s00259-018-4075-3). It would have been of added value to have an indication of possible underlying tauopathy in this case. Does dysartheria often associate with PSP/CBD tauopathy at postmortem assessment?
Receptive abilities seemed to be impaired as well based on Table 1: could this case then perhaps be classified as mixed dementia (cfr Mesulam) ?
Involvement of ACC is said to be related to attentional control, but I assume this could also explain the consistent degree of depressive symptoms observed in this patient. This is not discussed but affective change is also an important clinical symptom in cases with language production problems. Please elaborate.
Minor points:
Figure and table legends can be more extensive, indicating the statistics used, thresholds, color-scales, indication of left and right hemisphere and make use of abbreviation lists.
The English needs to be improved.
Reviewer 3 Report
Comments to the Author:
I have reviewed the manuscript entitled “Progressive dysarthria: a longitudinal single case study in an elder subject”. The overall contents of this manuscript are well organized to give a clear overview of this study. It is an interesting case report. I have some major comments as following:
-Authors should revise the introduction clearly including background and research gap between previous studies about dysarthria. The current form of the introduction is very short.
-Authors should add one more section related to “Material and Method”. In current version of manuscript, both results and method are mixed together. It should be written separately.
-Authors should add one more Figure about the flowchart of the PET and other preprocessing steps.
-What is the application of these findings in clinical research/neural disorder patients?
-Authors should write conclusion clearly after the discussion section.
Round 2
Reviewer 2 Report
The authors have answered most questions. While the main focus of this work is the speech profile, the availability of PET data is very valuable for further clinical interpretations. Hence, I insist to report the actual processing of PET data in more detail, since this is currently not clear. Please mention at least the time period: 90min-120min? And whether SUM PET or SUVR is shown in the figure.
An example of how to report PET processing: "The tracer was injected as a bolus in an antecubital vein (mean activity 150 MBq, SD 5 MBq, range
134–162 MBq). Scan acquisition started 90 min after tracer injection and lasted for 30 min. Prior to PET acquisition, a low-dose CT scan of the head was performed for attenuation correction. Random and scatter correction were applied. Data were recorded in list mode and reconstructed into six 5-min frames using ordered subsets expectation maximization (four iterations × 16 subsets). Processing of 18F-flutemetamol PET was done using SPM8 running on Matlab R2012b.."
Author Response
Please see the attachment, where you can find the file with the required changes (highlighted in blue).
